# Transcriptome Analyses Revealed the Wax and Phenylpropanoid Biosynthesis Pathways Related to Disease Resistance in Rootstock-Grafted Cucumber

**DOI:** 10.3390/plants12162963

**Published:** 2023-08-16

**Authors:** Yidan Wang, Ruifang Cao, Lu Yang, Xiaoyu Duan, Can Zhang, Xuejing Yu, Xueling Ye

**Affiliations:** Collage of Horticulture, Shenyang Agricultural University, 120 Dongling Road Shenhe District, Shenyang 110866, China; wyd9923@163.com (Y.W.); crf3528961620@163.com (R.C.); yanglu2020hhxx@163.com (L.Y.); ddxy1992@163.com (X.D.); zc19990227@163.com (C.Z.); yxj529@stu.syau.edu.com (X.Y.)

**Keywords:** RNA-seq, pest and disease control, cucumber, biosynthetic pathway, grafted cucumber

## Abstract

Cucumbers (*Cucumis sativus* L.) are a global popular vegetable and are widely planted worldwide. However, cucumbers are susceptible to various infectious diseases such as Fusarium and Verticillium wilt, downy and powdery mildew, and bacterial soft rot, which results in substantial economic losses. Grafting is an effective approach widely used to control these diseases. The present study investigated the role of wax and the phenylpropanoid biosynthesis pathway in black-seed pumpkin rootstock-grafted cucumbers. Our results showed that grafted cucumbers had a significantly higher cuticular wax contents on the fruit surface than that of self-rooted cucumbers at all stages observed. A total of 1132 differently expressed genes (DEGs) were detected in grafted cucumbers compared with self-rooted cucumbers. Pathway enrichment analysis revealed that phenylpropanoid biosynthesis, phenylalanine metabolism, plant circadian rhythm, zeatin biosynthesis, and diterpenoid biosynthesis were significantly enriched. In this study, 1 and 13 genes involved in wax biosynthesis and the phenylpropanoid biosynthesis pathway, respectively, were up-regulated in grafted cucumbers. Our data indicated that the up-regulated genes in the wax and phenylpropanoid biosynthesis pathways may contribute to disease resistance in rootstock-grafted cucumbers, which provides promising targets for enhancing disease resistance in cucumbers by genetic manipulation.

## 1. Introduction

Cucumbers (*Cucumis sativus* L., 2n = 2x = 14), a member of the cucurbit family [1], are one of the most economically valuable and nutritious vegetable crops worldwide [2]. The root system of cucumbers are underdeveloped, and their ability to resist pests and diseases and abiotic stress are weak. Grafting plays a critical role in safeguarding cucumber crops from severe infections. Cucumbers are particularly susceptible to many plant viruses, including cucumber mosaic virus, watermelon mosaic virus, and squash mosaic virus. Rootstocks can also be used to increase resistance to other destructive pathogens, such as bacterial wilt, Fusarium wilt, and Plectosporium blight [3]. Additionally, grafting with rootstocks that are tolerant or resistant to root-knot nematodes (RKN) has been observed to reduce RKN infestation and subsequent damage to cucumbers and other plants [4,5,6]. Grafting resistant varieties of cucumber in conjunction with specific rootstocks can reduce the incidence of these diseases and provide a measure of security for growers [7]. Overall, grafting with rootstocks provides numerous benefits for cucumber production, including improved access to water, disease resistance, pest control, and uniform growth [8,9]. Utilizing rootstocks resistant to viruses, nematodes, and other pathogens can protect against devastating epidemics and safeguard yields [10,11,12].

Lipid wax, also known as cuticular or epicuticular wax, is a hydrophobic crystalline substance found in the epidermal layers of plants [13]. In terrestrial plants, lipid wax covers the surface of most organs and is typically gray-green or white in color [14]. Previous studies have shown that lipid wax plays a crucial role in plant growth and development by serving as a hydrophobic barrier for plant epidermal cells, and as a protective layer for primary stomatal surfaces, thus, protecting against biotic and abiotic stress. For example, lipid wax can help reduce water loss, resist UV radiation, mitigate mechanical damage, alleviate high and low temperatures, and mitigate drought stress, allowing the plant to adapt to different environments [15]. Additionally, it can affect the feeding behavior of herbivorous insects and resist invasion by pathogenic fungi and bacteria. Further, lipid wax plays an important role in stress resistance by maintaining the cleanliness of plant surfaces and preventing organ fusion [16]. Lipid wax can vary greatly in composition and ultrastructure, depending on the developmental stage, tissue type, and environmental stimuli. Primarily composed of cutin, it is a complex and dynamic structure tightly regulated by a network of genes [17,18,19]. Recent advances in molecular biology and biochemistry have allowed the identification of key enzymes and transcription factors involved in cutin and wax biosynthesis, and signal transduction pathways that respond to drought, heat, and pathogen attack [19,20]. Cuticular wax biosynthesis in plants is a complex process involving multiple enzymatic and non-enzymatic pathways. The biosynthetic precursors of cuticular wax are mainly derived from very-long-chain fatty acids (VLCFAs) synthesized in plastids from acetyl-CoA via a fatty acid elongase (FAE) complex [21]. Subsequently, VLCFAs are modified by a series of enzymes, including fatty acid desaturases, fatty acid hydroxylases, and alcohol-forming acyl-CoA reductases, leading to the formation of a diverse array of alkanes, ketones, alcohols, and esters that constitute cuticular wax [21,22,23]. Cuticular wax crystal formation depends on the complex interplay between the wax physicochemical properties and plant cuticle microenvironment, including temperature, humidity, and atmospheric pressure [24]. Crystal structure formation is facilitated by the presence of cuticular wax-associated proteins that regulate wax molecule aggregation and alignment [18]. Cucumber fruit wax powder is a substance that covers the surface of cucumbers and directly affects their stress resistance.

Grafted cucumbers are widely used for disease resistance in cucumber-based crop production [25,26]. Disease resistance is primarily attributed to the expression of pathogen-related proteins, including chitinases, glucanases, ribonucleases, and hydrolytic enzymes [27,28]. In addition, the plant exhibits significant pylprophenanoid biosynthesis, which is essential for the development of systemic resistance against microbial pathogens [29]. Phenylpropanoids are important secondary metabolites that can be processed into diverse products involved in acquired disease resistance [30,31]. Phenylpropanoid biosynthesis is important for the synthesis of many structural and functional compounds in plants. Phenylpropanoids are metabolized through the shikimate, benzoate, and p-coumarate pathways to produce several phenylpropanoids and other compounds, including lignin, flavonoids, and phytoalexins [32]. Previous studies have demonstrated that phenylpropanoid biosynthesis is enhanced when plants are subjected to biotic or abiotic stresses, and many of these compounds have been shown to contribute to plant resistance to pathogens and stress [31]. Understanding the phenylpropanoid biosynthesis pathway in rootstock-grafted cucumbers can provide valuable insights into strategies aimed at increasing crop resistance against potential pathogenic agents.

This study aimed to investigate the biosynthetic pathway and its role in the development of wax and disease resistance in rootstock-grafted cucumber plants. Through transcriptome analyses, we gained insights into the physiological and genetic changes associated with disease resistance in rootstock-grated cucumbers. In particular, the enzymes involved in the biosynthetic pathway and how they may be related to disease resistance were explored to understand the overall chemical environment influencing disease resistance in rootstock-grafted cucumber plants.

## 2. Results

### 2.1. Wax Quantitative and Scanning Electron Microscopy Observation

Differences in fruit brightness between self-rooted and grafted cucumbers at 3, 6, 9, 12, and 15 d after flowering were calculated using a HP 200 precise colorimeter (Table 1). Our results indicated that the fruits surface wax contents of grafted cucumbers were significantly higher than that of self-rooted cucumbers at all stages. The wax contents were two-times higher than those of the self-rooted cucumbers at the harvest stage. However, it was not easy to distinguish between the waxes of self-rooted and grafted cucumbers using photographs of the fruits and SEM images (Figure 1).

### 2.2. RNA-Seq and Quality

A total of 269.88 million raw sequencing reads were assessed for quality and subjected to data filtering, and 253.75 million clean reads were obtained for further analysis. The filtered clean reads were mapped to the cucumber reference genome [33] using the HISAT [34] and Bowtie [35]. The total mapping ratio of self-rooted cucumbers (SR) and grafted cucumbers (BG) ranged from 95.12% to 95.76% and 94.00% to 94.96%, respectively. The GC content of all six samples exceeded 43.99% and the average base mass of Q30 was >91.70%. Moreover, principle component analysis of six transcriptomics showed that three samples from self-rooted cucumber (SR) and grafted cucumber (BG) were located within the 95% confidence region, indicating that the transcript sequencing data were reliable and of high quality (Appendix A).

### 2.3. Functional Annotation of Differently Expressed Genes (DEGs)

A total of 1132 DEGs (523 up-regulated and 609 down-regulated) were detected in BG cucumbers compared to those in SR cucumbers (Figure 2). Gene Ontology (GO) term analysis was performed to examine the expression profiles of the identified DEGs. The up-regulated and down-regulated DEGs from BG/SR were classified into three categories based on their functional annotations, including biological processes, cellular components, and molecular functions. The top 10 GO terms categorized as biological processes, cellular components, and molecular functions are shown in Figure 3.

DEGs in the biological process category, polysaccharide metabolic process (GO:0005976), cellular glucan metabolic process (GO:0006073), glucan metabolic process (GO:0044042), cellular polysaccharide metabolic process (GO:0044264), and cellular carbohydrate metabolic process (GO:0044262) were upregulated in BG cucumber. Among the cellular component categories, GO terms related to cell wall, cell periphery, extracellular region, apoplast, and external encapsulating structure (GO:0005618, GO:0071944, GO:0005576, GO:0048046, and GO:0030312) were up-regulated in BG cucumber. Interestingly, the GO terms chromatin, DNA packaging complex, DNA-directed RNA polymerase II, holoenzyme, protein-DNA complex, and mitochondrial part were all down-regulated in BG cucumber. In the molecular function category, the majority of xyloglucan: xyloglucosyl transferase activity, DNA-binding transcription factor activity, transcription regulator activity, and glucosyltransferase activity-related DEGs were up-regulated in BG cucumber.

### 2.4. Kyoto Encyclopedia of Genes and Genomes (KEGG) Analysis of Identified DEGs in Different Groups

Pathway enrichment analysis was performed to elucidate the biological functions of the identified DEGs using KEGG analysis. Using transcriptome sequencing, 1132 DEGs were assigned to 86 KEGG pathways in the BG cucumbers compared with those in the SR cucumbers (Table 2). The top 20 KEGG pathways are shown in Figure 4, among which phenylpropanoid biosynthesis, phenylalanine metabolism, plant circadian rhythm, zeatin biosynthesis, and diterpenoid biosynthesis were identified as significantly enriched (Table 2). In the BG cucumbers, 13 and 9 genes were up-regulated and down-regulated, respectively, in the phenylpropanoid biosynthesis pathway. Moreover, nine and two genes were up-regulated and down-regulated, respectively, during phenylalanine metabolism. Five genes in the zeatin biosynthesis pathway were down-regulated.

### 2.5. Verification of DEGs Involved in Wax and Phenylpropanoid Biosynthesis Using Quantitative Real-Time PCR

DEGs involved in wax and phenylpropanoid biosynthesis with more than two-fold differences between BG and SR cucumbers were selected for further qRT-PCR analysis. The primers used for quantitative real-time PCR (qRT-PCR) are listed in Appendix A. Twelve days after blooming, fruit skin from SR and BG cucumbers was used for qRT-PCR analysis. The qRT-PCR results showed the same expression pattern as the RNA-Seq data (Figure 5). The wax biosynthesis-related gene *CsaV3_3G010290* was significantly upregulated in BG cucumbers compared to SR cucumbers; however, *CsaV3_7G034800* did not show any difference in gene expression between BG and SR cucumbers. In BG cucumbers, phenylpropanoid biosynthesis-related genes, the expression of, for example, *CsaV3_4G002300*, *CsaV3_4G002320*, *CsaV3_4G002330*, *CsaV3_6G039690*, and *CsaV3_6G039710*, were significantly up-regulated, whereas the expression of *CsaV3_4G023630*, *CsaV3_7G005720*, and *CsaV3_7G031610* were significantly down-regulated.

## 3. Discussion

Recent studies have investigated the relationship between the cucumber cuticular wax layer and disease resistance, in addition to the impact of grafting. The cuticular wax layer functions as a physical barrier that limits the entry of pathogens and acts as a reservoir of signals that trigger plant defense responses [36]. In cucumbers, this wax layer has been found to play a significant role in disease resistance. Cuticular wax, a complex mixture of very-long-chain fatty acids and their derivatives, acts as a barrier to prevent the entry and spread of pathogens [18,24,37]. Some studies have shown that the thickness and composition of this wax layer can directly affect the plant’s resistance to diseases [38,39]. For instance, a thicker wax layer or a specific composition of wax constituents may enhance the plant’s resistance to certain pathogens. Wax accumulation was significantly enhanced in *MdMYB30*-ectopic-expression in *Arabidopsis*. Moreover, *MdMYB30* contributes to increased resistance against Pst DC3000 in *Arabidopsis*, thus indicating that epicuticular wax content and composition might enhance disease resistance against the bacterial pathogen Pst DC3000 [40]. Cuticle wax biosynthesis genes can regulate plant disease resistance by directly targeting defense-related genes, as reported by Wang [36]. For example, cucumber plants with *CsWAX2* overexpression showed a significant improvement in resistance to pathogens [41]. This study aimed to investigate the observed significant wax accumulation differences between grafted cucumber fruit and self-rooted cucumbers, and explore the molecular mechanisms involved. Consistent with these findings, our results showed significant wax accumulation in grafted cucumber fruit compared to self-rooted cucumbers from 3 to 15 d after flowering (Table 1). Gene expression profiling indicated the up-regulation of key genes involved in wax biosynthesis and transport processes. Overall, these findings suggest that grafting modifies the wax accumulation patterns in cucumber fruit, with potential implications for improved resistance to abiotic and biotic stressors. Cucumbers grafted with black seed pumpkin rootstock offer higher fruit production and resistance to diseases, such as Fusarium wilt. Understanding the role of the cuticular wax layer and the impact of grafting could provide new avenues for improving disease resistance in cucumbers and potentially other crop plants as well. Further research is needed to elucidate the specific regulatory mechanisms underlying the observed wax accumulation differences and to explore their potential impact on cucumber fruit development and post-harvest traits.

The resistance to diseases in cucumber plants is an interesting area of study in plant physiology. It has been found that this resistance is mediated by the phenylpropanoid biosynthesis pathway, a vital biochemical pathway in plants that is responsible for the production of secondary metabolites such as flavonoids and phenylpropanoids [42]. These secondary metabolites play a critical role in plant defense against various pathogens. Flavonoids and phenylpropanoids, as part of the plant’s immune response, have antimicrobial properties that help in warding off disease-causing microorganisms. They act as a physical barrier, preventing the entry and spread of pathogens. In addition, they are involved in signal transduction pathways that activate the plant’s defense mechanisms [43,44]. Phenylpropanoids also contribute to the structural integrity of the plant cell walls, making them more resistant to invasion by pathogens. Furthermore, they are involved in the production of lignin, a complex organic polymer that strengthens plant cell walls and is indigestible by most organisms, thereby providing an additional level of defense.

Recently, transcriptomics has become an essential tool for deciphering the molecular mechanisms underlying disease resistance in cucumber. Wu [45] used transcriptomic sequencing to examine the expression profiles of cucumber plants grown under low-temperature conditions. They found that low-temperature stress regulated the transcript levels of genes related to phenylalanine metabolism, the phytohormone ethylene, photosynthesis, flavonoid accumulation, lignin synthesis, cells, cycleactive gibberellin, and salicylic acid (SA) signaling in cucumber seedlings. Yadav [46] employed transcriptome sequencing to identify genes related to phytohormones and found that the phenylpropanoid pathway was involved in resistance to powdery mildew disease in watermelon. Genes related to the phenylpropanoid pathway were transcriptionally up-regulated in resistant cucumber lines. Moreover, transcriptome analysis showed that ethylene signaling pathways played a positive role in regulating the defense response of cucumbers to *Fusarium oxysporum* infection, as reported by Dong [47].

In this study, high-quality and reliable RNA-seq was performed on self-rooted and grafted cucumbers with black seed pumpkin rootstocks. Functional annotation of DEGs and KEGG analysis identified 13 and 9 genes that were up-regulated and down-regulated, respectively, in grafted cucumbers in the phenylpropanoid biosynthesis pathway. Recently, the phenylpropanoid biosynthesis pathway has been shown to play an important role in the mediation of cucumber disease resistance. Studies have reported that this pathway produces a range of secondary metabolites, such as lignin and flavonoids, which contribute to plant defense responses. For instance, Zhang [48] identified the involvement of the phenylpropanoid pathway in cucumber resistance to powdery mildew using metabolomic analysis. Previous studies have shown that the phenylpropane secondary metabolism pathway affects the synthesis of flavonoids and lignin, which can enhance stress tolerance [49,50].

Further investigations provided more detailed insights into the roles of specific genes related to the phenylpropanoid pathway in cucumber disease resistance. For example, Li [51] found that the overexpression of *CsMYB60*, a MYB transcription factor that regulates the expression of phenylpropanoid pathway genes, enhanced cucumber resistance to disease. The phenylpropanoid biosynthesis pathway in cucumber plants is a crucial component of their disease resistance strategy. Understanding this pathway can provide valuable insights into improving disease resistance in other crop plants as well, potentially leading to improved crop yields and food security. Future research in this area could focus on ways to enhance the activity of this pathway, thereby boosting the plant’s natural defenses against pathogens.

## 4. Materials and Methods

### 4.1. Plant Materials and Treatments

The cucumber variety ‘Zhongnong 18′ was used as the grafted scion, and a blackseed pumpkin variety was used as the rootstock. The rootstock blackseed pumpkin seeds were sown 5 d earlier than those of the ‘Zhongnong 18′ variety. When the scion cotyledons were unfolded, grafting was carried out using the insertion method. Both rootstock and grafting scions were grown in a 32-well tray and maintained in a greenhouse at the Shenyang Agricultural University in August 2021. Cucumber self-rooted seedlings were used as the control and black seed pumpkin rootstock grafting seedlings were used as the experimental treatment. In total, there were 20 plants per treatment with 3 replicates that were randomly arranged. Protective rows were set up around for daily management. The abbreviations ‘BG’ and ‘SR’ represent grafted and self-rooted populations.

### 4.2. Fruits Surface Wax Quantitative and Scanning Electron Microscopy Observation

The wax content of fruit skin from self-rooted and grafted cucumbers were measured using an HP-200 portable tri-stimulus colorimeter (Puxi Corp., Shanghai, China). The colorimeter was standardized using rectification plates before measurement. The ΔL value was used to represent the wax content on the fruit surface [52]. Three replicates of each treatment were used. The wax morphology and structure of different fruit samples were observed using scanning electron microscopy (SEM, HITACHI TM3030). The samples were prepared as described by Pang [53]. Starting from the flowering of female flowers, samples were taken from cucumber fruits with similar sizes and nodes. The sampling time was 3 days after flowering. On days 9, 12, and 15, 5 fruits were taken from each group and repeated 3 times to determine the amount of wax powder, and each sample was repeated 3 times.

### 4.3. RNA Isolation and Sequencing

Fruit skin from the self-rooted and grafted cucumbers (12 d after blooming) were used for RNA extraction. Total RNA was extracted using the RNeasy Plant Mini Kit (Qiagen), assessed using the RNA Nano 6000 Assay Kit of the Bioanalyzer 2100 system (Agilent Technologies, Santa Clara, CA, USA), and the RNA was sent to Beijing Novogene Bioinformatics Technology Company (Beijing, China) for sequencing and bioinformatics analysis. The Illumina NEBNext UltraTM RNA Library Prep Kit (Illumina, San Diego, CA, USA) was used for library construction. To select cDNA fragments 370–420 bp in length, the library fragments were purified using the AMPure XP system (Beckman Coulter, Beverly, MA, USA). PCR was performed using the Phusion High-Fidelity DNA polymerase, Universal PCR primers, and an Index (X) Primer. After PCR amplification and purification of the PCR products, the library quality was assessed using an Agilent Bioanalyzer 2100. After cluster generation using the TruSeq PE Cluster Kit version 4-cBot-HS (Illumina, San Diego, CA, USA), the library preparations were sequenced on an Illumina Novaseq platform, and 150 bp paired-end reads were generated.

### 4.4. RNA Seq Data Analysis

Clean data (clean reads) were obtained by removing reads containing adapters, poly-N, and low-quality reads from raw data. All downstream analyses were based on clean, high-quality data. Feature Counts (version 1.5.0-p3) was used to count the number of reads mapped to each gene. Subsequently, the Fragments Per Kilobase of transcript per Million mapped reads (FPKM) of each gene was calculated based on the length of the gene and the read count mapped to the gene. Differential expression analysis was performed in R using the DESeq2R package (version 1.20.0). All DEGs were functionally annotated using the KEGG, GO, Karyotic Orthologous Groups, PfAM, Swiss-Prot, TrEMBL, and NR databases.

### 4.5. Quantitative Real-Time PCR Validation of DEG Results

First-strand cDNA was prepared using SuperScript III Reverse Transcriptase (Invitrogen). qRT-PCR was performed as described previously [54] using SYBR Green Supermix (Bio-Rad Company, Hercules, CA, USA) with the CFX96™ Real-Time System. The Ct value for each gene was normalized to that of cs-actin. To determine the relative expression fold differences for each gene during different treatments, the formula 2^−∆∆Ct^ was calculated [55]. Gene-specific primers for quantitative real-time PCR were designed using the Primer 3 online program.

### 4.6. Data Statistical Method

This experiment was plotted using Excel, Origin 2022 software, NovoMagic (accessed on 20 August 2022) cloud platform, and SPSS 19.0 software for difference significance analysis. To test whether the data differences were significant, Duncan’s least significant range test (*p* = 0.05) was used to separate the means.

## 5. Conclusions

The phenylpropanoid biosynthesis pathway is a rich source of plant metabolites, necessary for lignin biosynthesis, and also the starting point for the production of many other important compounds. The phenylpropanoid biosynthesis pathway also plays a critical role in the response of cucumber to various pathogens. The identification of specific genes involved in this pathway provides promising targets for genetic manipulation to enhance disease resistance in cucumber. However, further studies are required to elucidate the intricate molecular mechanisms involved in the phenylpropanoid pathway and explore additional potential targets for manipulating cucumber disease resistance.

## Figures and Tables

**Figure 1 plants-12-02963-f001:**
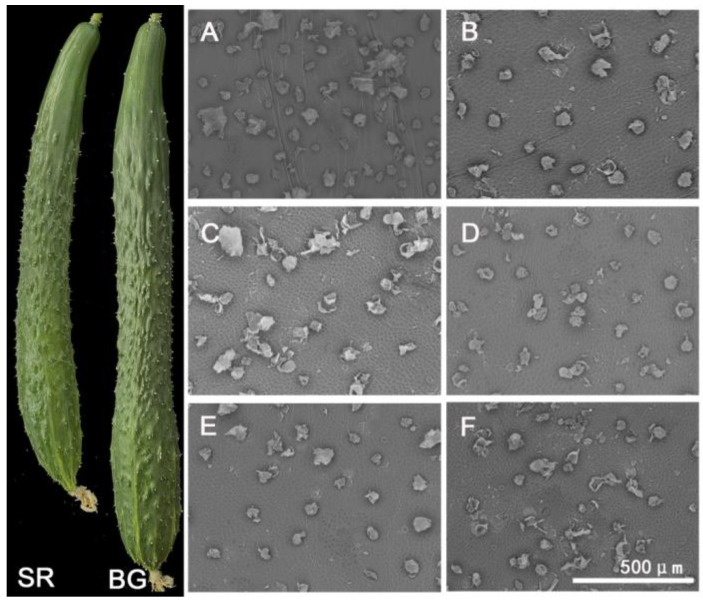
The fruits phenotype and wax layer of fruit skin from self-rooted and grafted cucumbers. SR: self-rooted cucumber; BG: grafted cucumber with black seed pumpkin rootstock; (**A**) wax layer of fruit skin from 9 d after flowering of self-rooted cucumber; (**B**) wax layer of fruit skin from 9 d after flowering of grafted cucumber; (**C**) wax layer of fruit skin from 12 d after flowering of self-rooted cucumber; (**D**) wax layer of fruit skin from 12 d after flowering of grafted cucumber; (**E**) wax layer of fruit skin from 15 d after flowering of self-rooted cucumber; (**F**) wax layer of fruit skin from 15 d after flowering of grafted cucumber.

**Figure 2 plants-12-02963-f002:**
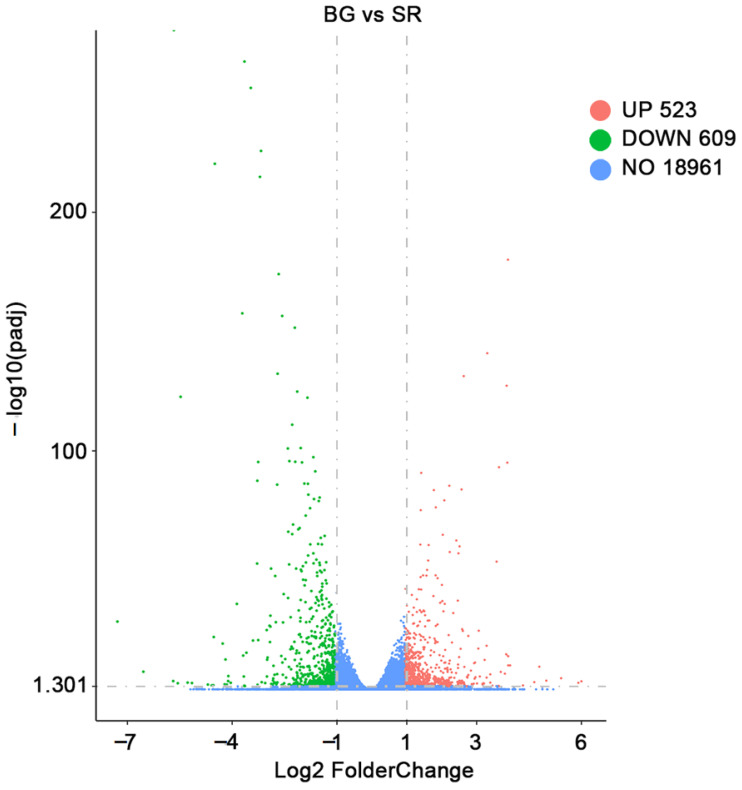
Volcano plot of differentially expressed genes (DEGs) of self-rooted cucumber (SR) and grafted cucumber with black seed pumpkin rootstock (BG).

**Figure 3 plants-12-02963-f003:**
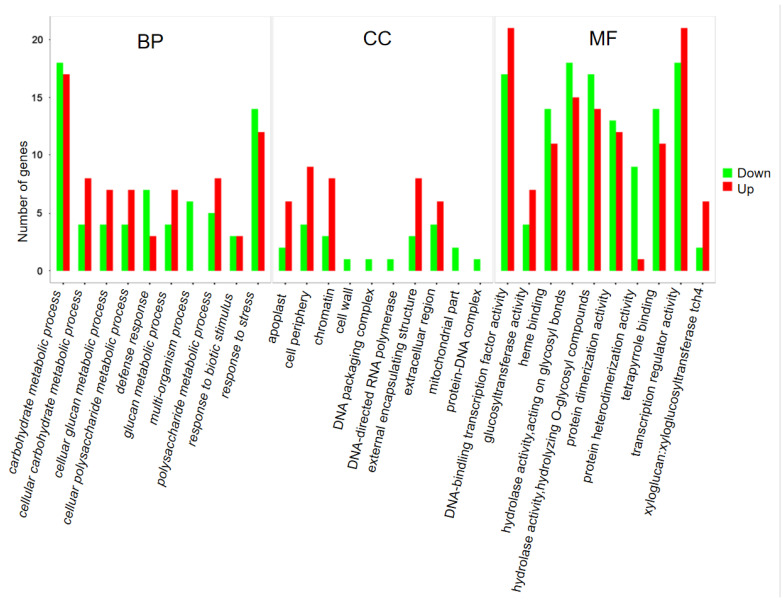
Functional classification and enrichment analysis of differentially expressed genes (DEGs) in the Gene Ontology (GO) term analysis. GO classification enriched significantly was analyzed with a criterion (*p* < 0.05). The DEGs were assigned into three classifications, including biological process (BP), cellular components (CCs), and molecular function (MF).

**Figure 4 plants-12-02963-f004:**
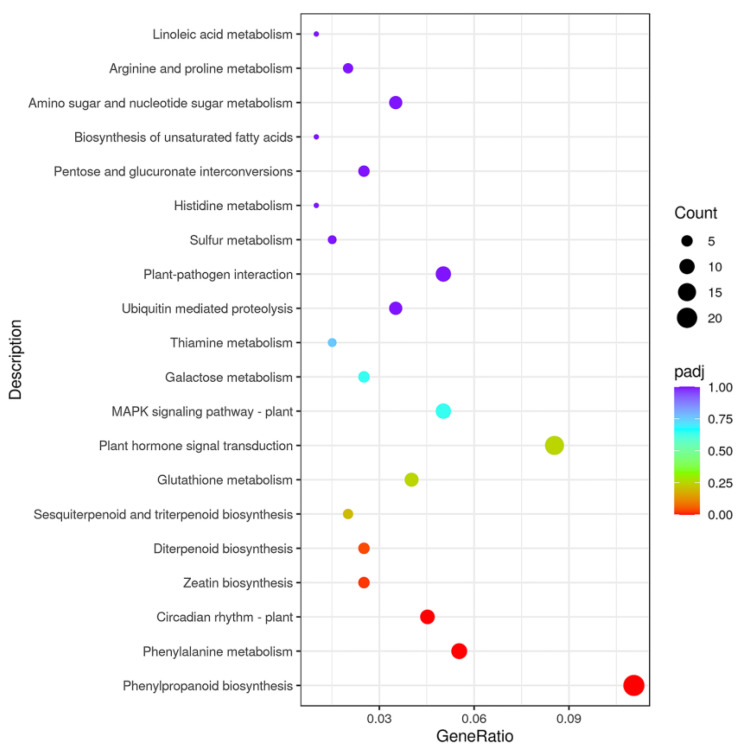
Kyoto Encyclopedia of Genes and Genomes (KEGG) enrichment analysis of differentially expressed genes (DEGs) for self-rooted (SR) and grafted (BG) cucumber. The top 20 pathways with highest enrichment level were exhibited according to the amount and enrichment level of DEGs annotated in BG cucumbers compared with those of SR cucumbers.

**Figure 5 plants-12-02963-f005:**
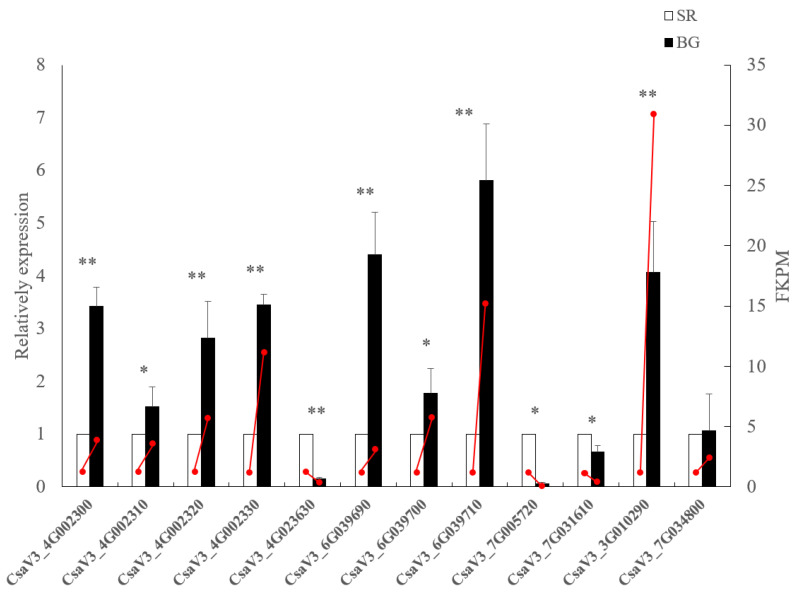
Relative expression of the differentially expressed genes (DEGs) involved in wax biosynthesis and phenylpropanoid biosynthesis in self-rooted cucumber (SR) and cucumber grafted with black seed pumpkin rootstock (BG). Red line represents FPKM value from RNA-seq. The bar chart and line graph display the relative gene expression results obtained from qRT-PCR and FPKM value from RNA-seq, respectively. Statistical significance: * *p* < 0.05, ** *p* < 0.01.

**Table 1 plants-12-02963-t001:** Average difference in surface brightness between self-rooted cucumber (SR) and grafted cucumber with black seed pumpkin rootstock (BG) at different sampling stages.

	Brightness/d				
Treatments	3 d	6 d	9 d	12 d	15 d
SR	1.03 ± 0.02 d	1.12 ± 0.02 d	1.24 ± 0.01 d	1.35 ± 0.12 d	1.23 ± 0.06 d
BG	1.31 ± 0.04 d **	1.53 ± 0.18 d *	1.88 ± 0.03 d **	2.87 ± 0.02 d **	2.39 ± 0.10 d **

Note: * represents *p* < 0.05, ** represents *p* < 0.01.

**Table 2 plants-12-02963-t002:** Analysis of significantly enriched KEGG pathways in grafted (BG) cucumber compared with self-rooted (SR) cucumber.

KEGG ID	Description	*p*-Value	*p*adj (Adjusted *p*-Value)	Up-Regulated Genes	Down-Regulated Genes
csv00940	Phenylpropanoid biosynthesis	2.14 × 10^−7^	1.84 × 10^−5^	*CsaV3_6G039710* *CsaV3_4G002330* *CsaV3_4G002310* *CsaV3_4G002320* *CsaV3_6G039680* *CsaV3_6G039690* *CsaV3_4G002290* *CsaV3_6G039700* *CsaV3_4G002300* *CsaV3_4G005430* *CsaV3_4G033920* *CsaV3_3G032830* *CsaV3_1G012650*	*CsaV3_4G023630* *CsaV3_6G043930* *CsaV3_4G023640* *CsaV3_2G018020* *CsaV3_6G039660* *CsaV3_7G031610* *CsaV3_7G031620* *CsaV3_7G005720* *CsaV3_2G036090*
csv00360	Phenylalanine metabolism	4.87 × 10^−6^	0.000205961	*CsaV3_6G039710* *CsaV3_4G002330* *CsaV3_4G002310* *CsaV3_4G002320* *CsaV3_6G039680* *CsaV3_6G039690* *CsaV3_4G002290* *CsaV3_6G039700* *CsaV3_4G002300*	*CsaV3_5G003800* *CsaV3_6G039660*
csv04712	Circadian rhythm—plant	7.18 × 10^−6^	0.000205961	*CsaV3_6G008360* *CsaV3_3G050020*	*CsaV3_5G014370* *CsaV3_6G005020* *CsaV3_2G025750* *CsaV3_5G000090* *CsaV3_7G024490* *CsaV3_4G024430* *CsaV3_1G005680*
csv00908	Zeatin biosynthesis	0.000970456	0.020864805		*CsaV3_1G040500* *CsaV3_5G006200* *CsaV3_5G039210* *CsaV3_3G040790* *CsaV3_6G018570*
csv00904	Diterpenoid biosynthesis	0.002158979	0.037134438	*CsaV3_3G001940* *CsaV3_3G049400* *CsaV3_5G005560*	*CsaV3_4G007790* *CsaV3_1G011060*

## Data Availability

The datasets generated and analyzed during this study are available on reasonable request from the corresponding authors. The raw data of RNA sequencing have been deposited in the NCBI Sequence Read Archive (SRA) repository under the accession numbers.

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
