# Peer review of "Transcriptome Analyses Revealed the Wax and Phenylpropanoid Biosynthesis Pathways Related to Disease Resistance in Rootstock-Grafted Cucumber"

_plants, 2023, doi:10.3390/plants12162963_

Round 1
Reviewer 1 Report
Manuscript ID: plants-2528146-peer-review-v1
Manuscript Title: Transcriptome analyses revealed the wax and phenylpropanoid
biosynthesis pathway related to disease resistance in rootstock grafted cucumber.
The title and subject of the manuscript are very interesting from the methodological and practical point of view, suitable and adequate. The scientific content contributes to the space in which it develops.
The abstract of the paper is factual concrete, realistic, and understandable.
The introduction provides a good understanding of the subject and its importance, with a significant quantity of information. Theoretical and practical reasons for the experiments are very reasonable.
The materials and methods are written clearly and in detail, for the reader to understand.
The results were described and discussed nicely and accurately.
There are some minor corrections that I have noticed that may improve the standard of the manuscript in the attached file.
I recommend that this manuscript be published after following the corrections suggested in the attached file.
I hope my comments improve the quality of your manuscript
Best regards

Author Response
Thank you for the reviewer's suggestion. It was your suggestion that made the overall content of the article richer and the format more standardized. Concrete content please see the attachment. In addition, I have highlighted corrected contentin yellow in the manuscript.

Reviewer 2 Report
1. For table 1, the authors should add units.
2. Line 119, the “BR: grafting” should be “BG: grafting”? The authors should also carefully check all the manuscript, such as line 168 and so on.
3. For “HISAT [37]and”, one blank should be inserted behind “[37].
4. Line 167, “related DEGS were” should be “related DEGs were”.
5. For Table 2, the gene ID should be italic.
6. In my opinion, the figure 5 should be re-drawn. Based on the descriptions of authors, the units for RNA-seq should be FPKM, the relative expression from qRT-PCR was calculated based on the ∆∆CT method. In this case, two Y-axis should be used.
7. Line 236, “Yadav V [44] employed” should be “Yadav [44] employed”.
8. Line 241, the “Fusarium oxysporum” should be italic.
9. The discussion part should be improved, for example, the authors should discuss more about the 13 up-regulated genes based on their annotation or orthologs from other crops.
10. The authors should carefully check the contents from line 266 to 270.
11. Line 277, “to Pang W [51].” should be “to Pang [51].”.
12. For the paragraph from line 270 to 277, the authors should add the detail sampling methods and the developmental stage of cucumber fruits.
13. The authors should add the statistical methods in the “Materials and Methods” part.
14. Line 346, the deposited ID in the NCBI is missing.
Author Response
Thank you for your valuable suggestion. I have made revisions in the manuscript and highlighted them in blue. Specific response please see the attachment.
